# IMPROVING ROBUSTNESS OF SOFTMAX CORSS-ENTROPY LOSS VIA INFERENCE INFORMATION

## ABSTRACT

Adversarial examples easily mislead the vision systems based on deep neural networks (DNNs) trained with the softmax cross entropy (SCE) loss. Such a vulnerability of DNN comes from the fact that SCE drives DNNs to fit on the training samples, whereas the resultant feature distributions between the training and adversarial examples are unfortunately misaligned. Several state-of-the-arts start from improving the inter-class separability of training samples by modifying loss functions, where we argue that the adversarial examples are ignored and thus limited robustness to adversarial attacks is resulted. In this paper, we exploit inference region which inspires us to involve a margin-like inference information to SCE, resulting in a novel inference-softmax cross entropy (I-SCE) loss, which is intuitively appealing and interpretable. The inference information is a guarantee to both the inter-class separability and the improved generalization to adversarial examples, which is furthermore demonstrated under the min-max framework. Extensive experiments show that under strong adaptive attacks, the DNN models trained with the proposed I-SCE loss achieve superior performance and robustness over the state-of-the-arts.

## 1 INTRODUCTION

Although deep neural networks have achieved state-of-the-art performance on various tasks (Szegedy et al., 2015; Zagoruyko & Komodakis, 2016; He et al., 2015; Huang et al., 2016; Larsson et al., 2016), it is recently shown that adversarial examples by adding imperceptible disturbances are not hard to fool *well-trained* neural networks (Szegedy et al., 2014; Goodfellow et al., 2015), leading to malfunction in intelligent systems such as image classification (Goodfellow et al., 2015; Szegedy et al., 2014), natural language processing (Jia & Liang, 2017; Carlini & Wagner, 2018), and autonomous driving (Liu et al., 2019; Chernikova et al., 2019). The vulnerability to adversarial attacks indicates that the neural networks do not convey proper feature representations and may *overfit* on the training samples which are even of large amounts (Ilyas et al., 2019). A reason of this issue is about the loss function used in training. Take the softmax cross entropy (SCE) loss as an example, which is widely adopted in regressing probabilities and is a core building block for high performance. The neural networks trained with SCE are shown to be limited in robustness to input perturbation, hence being suboptimal in real applications where adversarial attacks exist (Carlini & Wagner, 2017; Goodfellow et al., 2015; Kurakin et al., 2017b; Moosavi-Dezfooli et al., 2016; Papernot et al., 2016a).

The above issue brings many attempts that optimize SCE to enhance the robustness and anti-attack properties of neural networks (Sun et al., 2014; Schroff et al., 2015; Wen et al., 2016; Wan et al., 2018; Pang et al., 2020). These methods follow the same principle that they minimize the losses to maximally fit the training examples. However, the adversarial examples have a misaligned distribution with the training data, meaning that the fitted models in training could be repellent to the adversarial data (Ilyas et al., 2019). In fact, given a well-trained model, the distribution difference between the training and adversarial data is a blind region to the model, where we term as the inference region. The samples in this region are expected to be generalizable by the well-trained model, which is not the case in existing methods, resulting in the vulnerability of neural networks (Szegedy et al., 2014). The reason why this region exists, according to our analyses, is that the model overfits on the training data even when large amounts of data is accessible in training and the adversarial data

is clearly absent. Hence, how to generalize to the samples in this region still remains unresolved. Unfortunately, the above methods fail to take this fact into consideration.

In this paper, we exploit the inference region between the distributions of training data and adversarial examples. This region guides us to develop an inference schema which imposes a margin-like inference information on the predicted logit of the network. Based on this, we propose an inference-softmax cross entropy (I-SCE) loss. In this loss, the inference information is intuitively regarded as an additive term imposed on the prediction, which is extremely easy to implement and appealing. We further show the robustness of I-SCE under the Min-Max framework. Under severe adversarial attacks, I-SCE still maintains high accuracy and robustness, and has better resistance. The experiments on MNIST and CIFAR10 demonstrate that the proposed loss produces improved effectiveness and robustness compared with the state-of-the-art methods.

## 2 RELATED WORK

Adversarial attacks exist widely in open environment, imposing critical robustness demand of neural networks to the security quality and the overall performance of systems. Therefore, how to design an anti-attack and robust neural network has attracted the interest of many researchers, which are briefly reviewed in this section.

**Adversarial attack:** Szegedy et al. (2014) first proposed the concept of adversarial examples and employed the L-BFGS method as the solver of a disturbed problem to mislead neural networks. Goodfellow et al. (2015) proposed the Fast Gradient Symbol Method (FGSM) to generate adversarial examples with a single gradient step. Before backpropagation, FGSM was used to perturb the input of the model, which was an early form of adversarial training. Moosavi-Dezfooli et al. (2016) proposed the DeepFool which calculated the minimal necessary disturbance and applied it to construct adversarial examples. By imposing the $\ell_2$ regularization to limit the disturbance scale, DeepFool achieved good performance. After this, Madry et al. (2018) proposed the projected gradient descent (PGD) attack which had a strong attack strength, and was used in adversarial training to improve robustness. Recently, Guo et al. (2019) developed a local searching-based technique to construct a numerical approximation of the gradient, which was then used to perturb a small part of the input image.

**Adversarial defense:** The features of adversarial examples could follow a different distribution from the clean training data, making the defense progress very difficult. Distillation temperature was used to stabilize the gradient during training, thereby reducing the sensitivity of the model to disturbances (Papernot et al., 2016b). Metzen et al. (2017) introduced a novel model to detect adversarial examples. Chen et al. (2017) injected annealing noise into the softmax function during training to alleviate the early saturation problem of softmax loss. Xie et al. (2018) proposed the use of random resizing and random padding on images for defense. Ross & Doshi-Velez (2018) and Yan et al. (2018) proposed to regularize the gradients during training to improve the model robustness. Farnia et al. (2019) used a spectral regularization as the gradient penalty which was combined with adversarial training to alleviate vulnerability. In addition, data augmentation (Zhang et al., 2018; Hendrycks et al., 2020) was a typical option to enhance the generalization ability of neural networks and to reduce the risk of overfitting on training data. However, this option could not completely solve the problem of adversarial attack which always generated new kinds of adversarial examples. As a top performer, the adversarial training (AT) achieved advanced robustness in different adversarial attack environments (Kurakin et al., 2017a; Miyato et al., 2017; Madry et al., 2018; Sinha et al., 2018; Najafi et al., 2019; Shafahi et al., 2019). By using extra adversarial examples, it enabled the model to learn more generalizable feature representations. The AT mechanism accepted various losses and regularizers, and was a powerful tool to resist attacks. Despite of this, AT might sacrifice the performance on clean input and was computationally expensive (Xie et al., 2019). Schmidt et al. (2018) showed that the sample complexity of robust learning might be much larger than standard learning.

**Robust loss functions:** Lots of studies have been conducted to improve the widely used SCE loss function, most of which focused on encouraging higher intra-class compactness and greater separation between classes. The comparing loss (Sun et al., 2014) and the triplet loss (Schroff et al., 2015) were proposed to improve the internal compactness of each class which, however, suffered from the slowed training process and the unstable convergence. Center loss (Wen et al., 2016) avoided

the problem of slow convergence and instability by minimizing the Euclidean distance between features and the corresponding class centers, but the resultant robustness was not satisfactory. Liu et al. (2016) converted the softmax loss to the cosine space, and proposed that the angular distance margin favoured high intra-class compactness and inter-class separability. Wan et al. (2018) proposed the large-margin Gaussian Mixture loss, which used the Gaussian mixture distribution to fit the training data and increased the distance between feature distributions of different classes. Pang et al. (2020) proposed the Max-Mahalanobis center (MMC) loss to induce dense feature regions, which encouraged the model to concentrate on learning ordered and compacted representations.

Different from the previous works which improve the loss function to better fit the data distribution, the proposed method (*i.e.* I-SCE) is a much simple and interpretable way to enable the neural networks to learn freely. Moreover, we advocate that I-SCE encourages the models to be more generalizable with respect to the adversarial data instead of being overfitting on the training data.

## 3 METHODS

In this section, we introduce the inference-softmax cross entropy loss by first presenting the definition of inference region, which motivates us to develop an inference schema.

### 3.1 INFERENCE REGION

Current neural networks tend to overfit on the by-hand clean training data which, however, cannot work out a robust model and instead, makes them vulnerable to adversarial attacks. We advocate that this scenario is caused by the misaligned distribution between the clean training data and the adversarial data, and overfitting prevents the model to be tolerant to input perturbations. The distribution difference is termed as *inference region*, which characterizes why adversarial examples are outliers to the neural networks trained on clean data. Given Figure 1 as an illustration, the features of adversarial examples reside outside the feature space of training examples, whereas the decision boundary specified by the well-trained model closely fit the training data area. Considering that the type of adversarial attack incrementally appear in real scenarios, the decision boundary in Figure 1(a) is not good enough to give the right prediction, even if several kinds of input perturbation are involved in training. Instead, adversarial attacks are assumed to result in an isotropic expansion of the feature space, where the expanded region is the inference region as shown in Figure 1(b). Then, our task is to encourage the model generalizable to this region.

The softmax cross entropy (SCE) loss is a typical loss function used in training deep models, which imposes a hard constraint on the label of the input, *i.e.* regressing the probability of 1 on the correct label and the probability of 0 on the incorrect labels (generally in the case of one-hot label representation). Unfortunately, the hard constraint, on one hand, causes a difficult regression process in training, and on the other hand, makes the resultant model over-confident on the predictions, hence bringing the issue of vulnerability. This has already been mentioned in the literatures of label smoothing (Szegedy et al., 2016; Papernot et al., 2016b; Müller et al., 2019; Pereyra et al., 2017; Zou et al., 2019), which solves the problem by designing a soft label or a soft output distribution, *i.e.* regressing the probabilities of $1 - \epsilon$ and $\epsilon/K$ on the correct and incorrect labels, respectively, where $K$ is the number of task labels. We give an intuitive explanation of the above discussion in Figure 2(a) and Figure 2(b). As seen, the SCE encourages the label regression from one side along the 0-1 axis, whereas the label smoothing drives the regression from both sides around the target probabilities.

Besides, we also identify that the margin-based idea in SCE is similar to the label smoothing. Specifically, the soft label implies a margin between the true distribution and the soft output distribution. Considering that softmax is a monotonically increasing function, a margin between the label distributions can induce a margin between features in the logit layer of a neural network, as in the ArcFace loss (Deng et al., 2019). From Figure 2(c), we see that ArcFace pushes regression towards the target angles from both sides in a circle axis.

While the above analyses inform us that the regression is performed from either one side or both sides, here, we propose an alternative definition of soft label which could be regressed from arbitrary directions in feature space. Specifically, we free the circle constraint in ArcFace and impose the additive margin to the features only normalized by L2-norm. In this way, the resultant features are

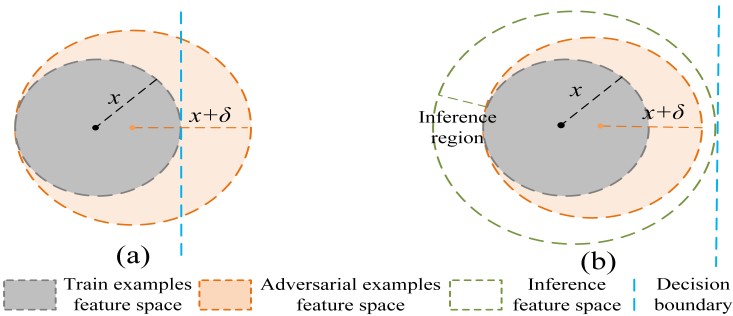

Figure 1: Illustration of the inference region: The grey circle region contains the features of the clean data $x$, and the orange circle region contains the features of the adversarial data $x + \delta$, where $\delta$ is the adversarial perturbation. When using SCE, the optimized decision boundary is located closely to the clean data area as shown in subfigure (a), whereas the expected boundary is around the adversarial data area as shown in subfigure (b). Considering the isotropic expansion of the space caused by adversarial perturbation, the inference region is then induced from the annular area.

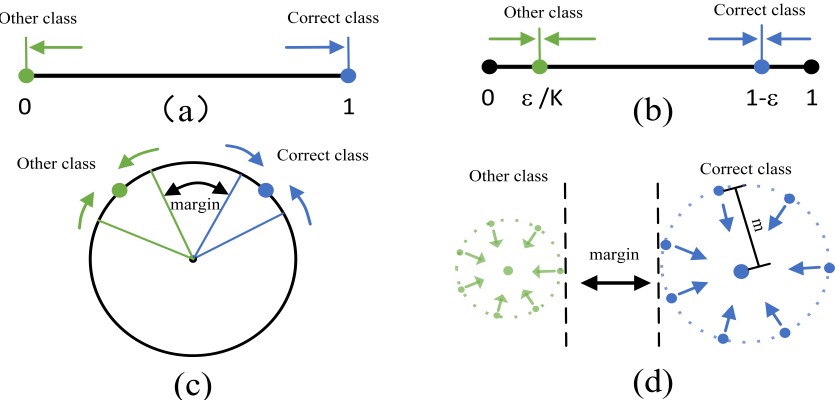

Figure 2: Intuitive explanation of label regression. (a) is the softmax cross entropy case Which regresses the probabilities from one side. (b) is the label smooth case which regresses the soft labels from both sides. (c) is the ArcFace case which regresses the targets on a circle axis in feature space, *i.e.* encouraging the circular margins between different classses. (d) is the inference softmax cross entropy case which regresses the targets from all directions, *i.e.* encouraging the isotropic margins between different classes.

not necessarily located on a circle or a sphere, and on the contrary, the marge is isotropically posed around each sample in the feature space, as shown in Figure 2(d). We will empirically demonstrate the effectiveness of this operation over the ArcFace.

By implementing this margin idea, the inference information is then contained in the margin, which could help 1) to avoid overfitting and 2) to improve the generalization ability of the feature representation, driving the decision boundary towards the boundary of inference region in feature space. Hence, the small perturbation of an adversarial examples is not easy to cross the decision boundary, greatly alleviating the problem of vulnerability. This schema is simple, interpretable, and effective as demonstrated in experiments. In the following, we present the inference-softmax cross entropy in details.

## 3.2 INFERENCE-SOFTMAX CROSS ENTROPY

To derive a robust loss for neural network training, in this section, we apply the inference-schema on SCE and propose an inference-softmax cross entropy (I-SCE) loss, which could encourage the tolerance of the model to adversarial perturbations, thus avoid overfitting.

Given a $k$-class classification task, the posterior probability predicted by the deep model using softmax is

$$P(y' = i|x) = \frac{e^{f_i(x)}}{\sum_j e^{f_j(x)}}, \tag{1}$$

where $i \in [1, k]$ is the label candidate, and $f_i$ is the prediction function for the $i$-th class which specifies both the backbone and the softmax layer in a typical classification network. To improve the vulnerability of SCE, we impose the inference information to the logits produced by the neural networks and propose an inference softmax as

$$P_I(y' = i|x) = \frac{e^{sf_i(x)+m}}{e^{sf_i(x)+m} + \sum_{j \neq i} e^{f_j(x)}}, \tag{2}$$

which then induces the inference-softmax cross entropy loss as

$$\text{I-SCE} = -\sum_{i=1}^{k} y_i \ln \frac{e^{y_i(sf_i(x)+m)+(1-y_i)f_i(x)}}{e^{y_i(sf_i(x)+m)+(1-y_i)f_i(x)} + \sum_{j \neq i} e^{f_j(x)}}, \tag{3}$$

where $y_i = 1$ if the ground truth label of $x$ is $i$ and otherwise 0, and $s \geq 1$ is used to scale the predication $f_i(x)$ and control the gradient update rate on the right class. Note that we use $y_i$ as an indicator of the inference information, that is, $s$ and $m$ are only imposed on the right class instead of all classes. As seen, the implementation of this loss is very easy by simply adding a scalar and a constant on the prediction of the right class, which is unaggressive to the original training code of neural networks.

In the implementation of I-SCE, we find that the case of $f_i \gg f_j, j \neq i$ possibly occurs, which reduces the effect of $m$. To address this issue, we normalize $f(x)$ by $L_2$ to increase the numerical stabilization. During the inference process, Eq. 2 is calculated by firstly finding the index $i$ of the maximal value $f_i(x)$ among $i \in [1, k]$ and then applying $s$ and $m$ on the $i$-th class according to this equation. This operation does not change the class decision since $s \geq 1$ and $m > 0$.

### 3.3 ROBUSTNESS ANALYSIS OF I-SCE

#### 3.3.1 EXPECTED INTERVAL OF CORRECT CLASS

To demonstrate the robustness of I-SCE, we analyze the expected intervals of the correct class predicted by both I-SCE and SCE. Here, assume the minimum perturbation $\delta$ which makes the model just misclassified. The probability that the SCE model recognizes the adversarial sample $x + \delta$ as the correct label $i$ is

$$P(i|x + \delta) = \frac{e^{f_i(x+\delta)}}{\sum_j e^{f_j(x+\delta)}}. \tag{4}$$

Regarding the I-SCE model, the probability is then

$$P_I(i|x + \delta) = \frac{e^{sf_i(x+\delta)+m}}{e^{sf_i(x+\delta)+m} + \sum_{j \neq i} e^{f_j(x+\delta)}}. \tag{5}$$

The expected intervals of the correct class by using SCE and I-SCE are defined as

$$L = P(i|x) - P(i|x + \delta) = \frac{e^{f_i(x)}}{\sum_j e^{f_j(x)}} - \frac{e^{f_i(x+\delta)}}{\sum_j e^{f_j(x+\delta)}} \tag{6}$$

and

$$\begin{aligned} L_I &= P_I(i|x) - P_I(i|x + \delta) \\ &= \frac{e^{sf_i(x)+m}}{e^{sf_i(x)+m} + \sum_{j \neq i} e^{f_j(x)}} - \frac{e^{sf_i(x+\delta)+m}}{e^{sf_i(x+\delta)+m} + \sum_{j \neq i} e^{f_j(x+\delta)}}, \end{aligned} \tag{7}$$

respectively. The vulnerability of SCE to adversarial attacks states that $f(x + \delta) < f(x)$. Considering that the perturbation $\delta$ is a just value that misleads the SCE model, the expected interval measures the maximal level of perturbation that the model is robust on. The larger the interval is, the more robust the model is. Starting from this point, we show the following property of I-SCE:

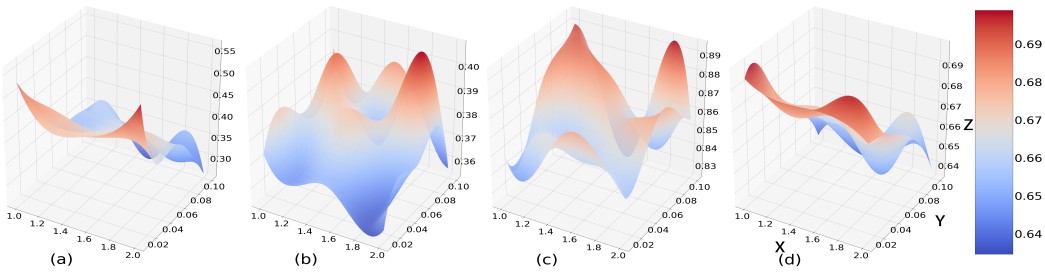

Figure 3: Performance of I-SCE under different parameter settings. The x-axis is $s$, the y-axis is $m$, and the z-axis is the accuracy. (a) Deepfool attack on MNIST. (b) Deepfool attack on CIFAR10. (c) Untargeted PGD attack on MNIST. (d) Untargeted PGD attack on CIFAR10.

*When $s \geq 1$, $m > 0$, and $\frac{se^{sf_i(x)+m}}{(e^{sf_i(x)+m}+\sum_{j\neq i} e^{f_j(x)})^2} - \frac{e^{f_i(x)}}{(\sum_j e^{f_j(x)})^2} > 0$, $L_I$ is larger than $L$.*

The condition in the above property is both theoretically demonstrated and empirically validated in Appendix A.2. This states that the robustness of I-SCE is improved compared with SCE.

### 3.3.2   MIN-MAX FRAMEWORK

The above robustness conclusion is also applicable to the Min-Max framework (Madry et al., 2018), which is a typical framework of adversarial attack and defense. The Min-Max framework is formulated as

$$\min_\theta \rho(\theta), \quad where \quad \rho(\theta) = \mathbb{E}_{(x,y)\sim D}\left[\max_{\delta\in S} \ell(\theta, x+\delta, y)\right], \tag{8}$$

where $\theta$ is the model parameter and $\delta$ is the input perturbation. The internal maximization is an attack process that finds the perturbation that maximally mislead the model $\theta$. The external minimization is a defense process that encourages the model tolerant to such an attack. We use $\rho_I$ and $\rho$ to represent the objective losses by using I-SCE and SCE, respectively. Given an input perturbation $\delta$ and a trained model $\{f_i\}$, we have *when $s \geq 1$ and $m > 0$,*

$$P_I(i|x+\delta) > P(i|x+\delta), \tag{9}$$

which is proven in Eq. 12 of Appendix A.2. This states that the $P_I$ results in a lower loss than $P$, *i.e.* $\rho_I < \rho$. Hence, the lower loss indicates the better defense performance on adversarial attacks, which demonstrates the improved robustness of I-SCE.

### 3.4   RELATIONSHIP WITH LARGE MARGIN LEARNING

While the proposed method (I-SCE) could be viewed as a margin-based loss, the difference to the ArcFace loss is how the margin (or inference information) is applied to the logits. ArcFace loss normalizes the features and the weights such that the resultant features are located on a hypersphere, and the training process regresses the class targets along the surface of the hyper-sphere. Instead, the proposed method only normalizes the features but not the weights in order to locate the features in a free space, in which case the regression process can be performed in any direction. We advocate that freeing the sphere constraint will bring performance improvement of adversarial defencing, which is demonstrated in the experiments. The reason of the effectiveness may be that the adversarial perturbation causes a large variation in the feature space. Constraining the features on a hyper-sphere would bring a large feature shift if the normalization direction (to the sphere) is undesirable. By contrast, the proposed method prefers isotropic tolerance to feature perturbations, hence being better.

## 4   EXPERIMENTS

In this section, we conduct a series of experiments on MNIST (Lecun & Bottou, 1998) and CIFAR-10 (Krizhevsky & Hinton, 2009) to demonstrate the effectiveness of the proposed I-SCE. The backbone used in our implementation is ResNet-32 with five stages (He et al., 2016), which is optimized

Table 1: Classification accuracy (%) under PGD attack on MNIST.

| Method | Clean | $\epsilon = 0.02$ | | | | $\epsilon = 0.04$ | | | |
|---|---|---|---|---|---|---|---|---|---|
| | | $PGD_{10}^{un}$ | $PGD_{10}^{tar}$ | $PGD_{50}^{un}$ | $PGD_{50}^{tar}$ | $PGD_{10}^{un}$ | $PGD_{10}^{tar}$ | $PGD_{50}^{un}$ | $PGD_{50}^{tar}$ |
| SCE | 99.53 | 64.38 | 16.03 | 0.34 | 0.02 | 15.15 | 4.05 | 0.32 | 0.01 |
| Center | 99.46 | 39.29 | 14.55 | 3.16 | 4.18 | 16.64 | 2.90 | 0.61 | 0.49 |
| L-GM | 99.59 | 54.31 | 34.21 | 2.41 | 0.70 | 20.64 | 13.68 | 0.25 | 0.67 |
| ArcFace | 99.43 | 84.03 | 68.34 | 52.12 | 13.08 | 64.50 | 48.73 | 30.75 | 6.03 |
| MMC | 99.45 | 80.87 | 53.26 | 46.06 | 7.69 | 58.63 | 35.35 | 29.76 | 4.45 |
| I-SCE | **99.57** | **85.48** | **70.48** | **63.47** | **12.74** | **68.08** | **49.70** | **46.61** | **6.78** |
| Random | 99.30 | 65.10 | 20.40 | 0.34 | 0.14 | 15.64 | 14.25 | 0.32 | 0.12 |
| LS | 99.54 | 66.88 | 33.99 | 0.37 | 0.02 | 21.69 | 1.25 | 0.36 | 0.01 |
| AT | 99.25 | 97.73 | 98.86 | 13.62 | 6.73 | 90.59 | 94.44 | 0.74 | 0.01 |

Table 2: Classification accuracy (%) under PGD attack on CIFAR10.

| Method | Clean | $\epsilon = 0.01$ | | | | $\epsilon = 0.04$ | | | |
|---|---|---|---|---|---|---|---|---|---|
| | | $PGD_{5}^{un}$ | $PGD_{5}^{tar}$ | $PGD_{50}^{un}$ | $PGD_{50}^{tar}$ | $PGD_{5}^{un}$ | $PGD_{5}^{tar}$ | $PGD_{50}^{un}$ | $PGD_{50}^{tar}$ |
| SCE | **90.45** | 19.00 | 6.74 | 6.17 | 0.02 | 6.29 | 4.28 | 5.56 | 0.01 |
| Center | 88.98 | 25.09 | 5.33 | 4.55 | 1.07 | 10.82 | 3.94 | 3.08 | 0.98 |
| L-GM | 89.02 | 20.39 | 16.04 | 6.47 | 0.24 | 6.84 | 8.04 | 5.04 | 0.20 |
| ArcFace | 90.11 | 31.44 | 34.28 | 9.69 | 0.95 | 8.84 | 23.33 | 3.36 | 1.02 |
| MMC | 89.97 | 31.83 | 20.56 | 10.84 | 1.59 | 16.78 | 15.97 | 4.52 | 1.47 |
| I-SCE | 89.09 | **36.95** | **44.86** | **14.82** | **5.26** | **16.86** | **25.50** | **5.51** | **1.95** |
| Random | 86.09 | 21.29 | 57.12 | 7.51 | 28.75 | 7.17 | 20.77 | 5.62 | 0.46 |
| LS | 90.35 | 20.20 | 5.57 | 6.38 | 0.01 | 6.17 | 4.27 | 5.99 | 0.01 |
| AT | 83.48 | 56.30 | 77.95 | 7.87 | 0.08 | 25.36 | 24.05 | 7.15 | 0.07 |

by using the Adam algorithm (Kingma & Ba, 2015). We employ the white-box attack and the black-box attack including the targeted and untargeted PGD (Madry et al., 2018), deepfool (Moosavi-Dezfooli et al., 2016), and SimBA (Guo et al., 2019). We select the state-of-the-arts as competitors, such as the Center loss (Wen et al., 2016), the large-margin Gaussian Mixture (L-GM) loss (Wan et al., 2018), ArcFace loss (Deng et al., 2019), the Max-Mahalanobis center (MMC) loss (Pang et al., 2020), the random method (Xie et al., 2018), Label Smoothing(Szegedy et al., 2016), and the adversarial training (AT) method (Madry et al., 2018).

## 4.1 ABLATION STUDIES

There are two hyper-parameters $s$ and $m$ in the proposed I-SCE, which affects the defense performance. We set the ranges as $s \in [1, 2]$ and $m \in (0, 0.1]$, and densely evaluate the performance of I-SCE under different settings and different attacks. Figure 3 illustrates the results, from which we see that the performance is highly correlated with the settings, the attack types, and the datasets. Therefore, to get better robustness, the parameters need to be reset in different tasks by using a small validation set. In the following experiments, to make fair comparison, we set $s = 1$ and $m = 0.1$.

## 4.2 COMPARISON WITH STATE-OF-THE-ARTS

**PGD attack:** The PGD attack is a strong white-box untargeted and targeted attack. We use $L_2$ constrained untargeted and targeted PGD attacks for comparison. The results are listed in Table 1 and Table 2. The Clean column is the accuracy on clean samples, $\epsilon$ is the perturbation level, and $PGD_{10,50}^{tar,un}$ represents the targeted or untargeted attacks with 10 or 50 iterations. The results indicate that I-SCE produces better performance than the others in most cases. While AT sometimes achieves good performance, it has a noticeable sacrifice of accuracy on clean examples, *e.g.* on CIFAR10 and it has weaker defense against strong PGD attacks than I-SCE. By constrast, I-SCE preserves high

Table 3: Classification accuracy (%) under Deepfool attack.

| | | MNIST | | | | CIFAR10 | | | |
| | | $\epsilon = 0.01$ | | $\epsilon = 0.04$ | | | $\epsilon = 0.01$ | | $\epsilon = 0.04$ | |
| Method | Clean | $DF_{10}$ | $DF_{50}$ | $DF_{10}$ | $DF_{50}$ | Clean | $DF_3$ | $DF_{20}$ | $DF_3$ | $DF_{20}$ |
|---|---|---|---|---|---|---|---|---|---|---|
| SCE | 99.53 | 27.16 | 23.74 | 26.75 | 23.58 | **90.45** | 10.24 | 4.95 | 10.29 | 4.86 |
| Center | 99.46 | 7.58 | 3.58 | 7.23 | 3.34 | 88.98 | 10.06 | 8.01 | 9.85 | 7.96 |
| L-GM | 99.59 | 3.81 | 1.92 | 3.26 | 1.76 | 89.02 | 17.98 | 6.63 | 17.34 | 6.58 |
| ArcFace | 99.43 | 74.04 | 64.35 | 73.09 | 64.05 | 90.11 | 62.45 | 54.90 | 62.24 | 54.01 |
| MMC | 99.45 | 2.46 | 3.07 | 2.42 | 3.05 | 89.97 | 4.28 | 2.57 | 4.12 | 2.48 |
| I-SCE | **99.60** | **90.30** | **78.21** | **89.24** | **77.78** | 89.09 | **67.50** | **60.78** | **67.14** | **60.74** |
| Random | 99.30 | 27.03 | 24.30 | 27.12 | 22.99 | 84.15 | 21.09 | 14.41 | 18.09 | 5.69 |
| LS | 99.54 | 27.24 | 17.77 | 26.72 | 17.62 | 90.35 | 10.03 | 5.05 | 9.63 | 4.97 |
| AT | 99.25 | 32.21 | 5.10 | 31.50 | 4.51 | 83.48 | 47.60 | 23.10 | 46.40 | 22.39 |

Table 4: Classification accuracy (%) under SimBA attack.

| | | MNIST | | | | CIFAR10 | | |
| Method | Clean | $\epsilon = 0.5$ | $\epsilon = 1$ | $\epsilon = 1.5$ | Clean | $\epsilon = 0.5$ | $\epsilon = 1$ | $\epsilon = 1.5$ |
|---|---|---|---|---|---|---|---|---|
| SCE | 99.33 | 97.00 | 91.23 | 88.43 | **90.17** | 77.60 | 72.77 | 70.17 |
| Center | 99.25 | 96.20 | 91.04 | 88.17 | 89.27 | 81.80 | 77.94 | 76.48 |
| MMC | 99.18 | 98.38 | 97.42 | 96.15 | 89.77 | 82.57 | 77.83 | 76.17 |
| I-SCE | **99.40** | **98.73** | **97.90** | **97.10** | 89.63 | **85.93** | **83.33** | **82.07** |
| Random | 98.90 | 94.84 | 92.32 | 86.12 | 89.48 | 76.54 | 72.74 | 71.30 |
| AT | 98.86 | 97.32 | 66.98 | 49.94 | 83.46 | 80.14 | 75.40 | 72.22 |

performance on clean data. Under several attack cases, *e.g.* $\epsilon = 0.04$, I-SCE performs better than the others and is comparable with MMC and AT.

**Deepfool attack:** The Deepfool attack generates minimal input perturbations to mislead the neural Networks. Here, we use the $L_2$ constrained Deepfool attack on MNIST and CIFAR10. From the results in Table 3, it is clearly observed that I-SCE produces much higher performance than all competitors, which have very limited defense ability against Deep fool. The performance improvement of I-SCE is above 50% in most cases, which is significant and exciting. In real applications, the minimal disturbance generated by Deepfool is more usual than the strong offensive disturbance generated by PGD. Therefore, the results indicate that I-SCE is more suitable and can achieve better performance in real scenarios than the other methods.

**Black-box attack:** Robust performance is critical to claim reliable robustness against the black-box attacks (Carlini et al., 2019). SimBA (Guo et al., 2019) is a black-box query-based attack, which is employed here. We set the frequency of query as 300 times per image on MNIST and 500 times per image on CIFAR10. The results under different disturbance levels are shown in Table 4, from which we see that I-SCE has higher accuracy and little sacrifice of accuracy compared with the others. This evidence indicates that I-SCE can induce reliable robustness rather than the false one caused by, *e.g.*, gradient mask (Athalye et al., 2018).

**Feature embedding:** To visually investigate the effect of I-SCE, we compute a 3d representation of the input by adding a three-dimensional embedding layer before the output layer. The embedded points are plotted in Figure 4, where the samples are selected from the test set of MNIST and CIFAR10 without any perturbation. As seen, the samples of SCE distribute confusedly in the space, where little perturbations on the samples could change the category decision. In contrast, I-SCE produces separable clusters for each classes with large margins among them, which hence has higher tolerance to the perturbations than the other competitors.

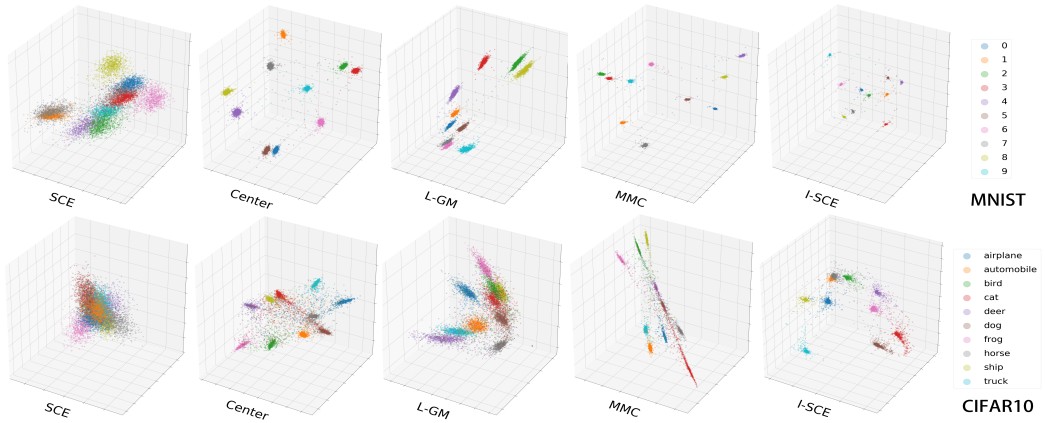

Figure 4: Illustration of three-dimensional feature embedding.

## 5 CONCLUSION

The original SCE loss induces the model to fit the distribution of the clean data, which is shown vulnerable to adversarial attacks. We advocate that the vulnerability is caused by the unawareness of the inference region during learning. Targeting at this issue, we propose an inference-SCE loss that avoids overfitting by imposing an additive inference information to the output of the neural network such that the sensitive class region of the model is expanded. In this way, the model has higher generalization ability to the adversarial examples. Extensive experiments demonstrate the superiority of I-SCE compared with the state-of-the-arts. Especially in the case of strong attacks, I-SCE still remains high robustness.

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

## A    APPENDIX

### A.1    EXPERIMENTS ON SHALLOW NETWORKS

In real applications on mobiles, shallow networks are generally preferred because of low computational costs. Hence, in this section, we evaluate the robustness of the proposed I-SCE with shallow networks. Specifically, we following the same settings of competitors and attack methods as in Section 4. The backbone network is LeNet-5 (Lecun & Bottou, 1998) for MNIST and an 8-layer neural network for CIFAR10.

Table 5 and Table 6 illustrate the performance under the PGD attack on MNIST and CIFAR10, respectively. The results indicate that I-SCE performs surprisingly well in all cases of attack, while remaining slight sacrifice of accuracy on clean data. Notably, the performance gaps between I-SCE and the others are above 50% in many cases, which validates the effectiveness of the proposed schema. More importantly, under severe attacks, I-SCE still shows strong robustness.

Table 7 lists the results of all methods under the Deepfool attack. We find that the performance of I-SCE is comparable with the state-of-the-arts. MMC produces the best accuracy under attacks, but has noticeable sacrifice of accuracy on clean data. By contrast, I-SCE shows better trade-off between accuracy and robustness.

### A.2    PROVE OF THE PROPERTY ON EXPECTED INTERVAL OF CORRECT CLASS

According to the definition $L$ and $L_I$ in Eq. 6 and Eq. 7, we can derive

$$
\begin{aligned}
L_I - L = {} & \frac{e^{sf_i(x)+m}}{e^{sf_i(x)+m} + \sum_{j\neq i} e^{f_j(x)}} - \frac{e^{sf_i(x+\delta)+m}}{e^{sf_i(x+\delta)+m} + \sum_{j\neq i} e^{f_j(x+\delta)}} \\
& - \frac{e^{f_i(x)}}{\sum_j e^{f_j(x)}} + \frac{e^{f_i(x+\delta)}}{\sum_j e^{f_j(x+\delta)}}.
\end{aligned}
\tag{10}
$$

By defining

$$
h(f(x)) = P_I(i|x) - P(i|x) = \frac{e^{sf_i(x)+m}}{e^{sf_i(x)+m} + \sum_{j\neq i} e^{f_j(x)}} - \frac{e^{f_i(x)}}{\sum_j e^{f_j(x)}},
\tag{11}
$$

we then have

$$
\begin{aligned}
h(f(x)) = P_I(i|x) - P(i|x) &= \frac{e^{sf_i(x)+m}}{e^{sf_i(x)+m} + \sum_{j\neq i} e^{f_j(x)}} - \frac{e^{f_i(x)}}{\sum_j e^{f_j(x)}} \\
&= \frac{e^{sf_i(x)+m} \sum_j e^{f_j(x)} - e^{f_i(x)}\left(e^{sf_i(x)+m} + \sum_{j\neq i} e^{f_j(x)}\right)}{\left(e^{sf_i(x)+m} + \sum_{j\neq i} e^{f_j(x)}\right)\sum_j e^{f_j(x)}} \\
&= \frac{e^{sf_i(x)+m} \sum_{j\neq i} e^{f_j(x)} - e^{f_i(x)} \sum_{j\neq i} e^{f_j(x)}}{\left(e^{sf_i(x)+m} + \sum_{j\neq i} e^{f_j(x)}\right)\sum_j e^{f_j(x)}} \\
&= \frac{\sum_{j\neq i} e^{f_j(x)}\left(e^{sf_i(x)+m} - e^{f_i(x)}\right)}{\left(e^{sf_i(x)+m} + \sum_{j\neq i} e^{f_j(x)}\right)\sum_j e^{f_j(x)}}.
\end{aligned}
\tag{12}
$$

Table 5: Performance (%) of shallow neural networks under PGD attack on MNIST.

| Method | Clean | $\epsilon = 0.02$ | | | | $\epsilon = 0.04$ | | | |
|--------|-------|-------------------|--|--|--|-------------------|--|--|--|
| | | $\text{PGD}_{10}^{un}$ | $\text{PGD}_{10}^{tar}$ | $\text{PGD}_{50}^{un}$ | $\text{PGD}_{50}^{tar}$ | $\text{PGD}_{10}^{un}$ | $\text{PGD}_{10}^{tar}$ | $\text{PGD}_{50}^{un}$ | $\text{PGD}_{50}^{tar}$ |
| SCE | **99.12** | 65.55 | 91.43 | 0.75 | 6.12 | 6.40 | 52.47 | 0.72 | 2.71 |
| Center | 98.17 | 74.44 | 96.65 | 1.21 | 0.43 | 8.97 | 72.04 | 1.20 | 0.30 |
| L-GM | 97.92 | 72.08 | 25.54 | 50.17 | 0.03 | 50.43 | 2.73 | 49.32 | 0.01 |
| MMC | 91.02 | 82.34 | 82.96 | 35.32 | 12.30 | 56.49 | 53.74 | 35.21 | 7.14 |
| I-SCE | 98.42 | **98.74** | **98.40** | **98.55** | **95.28** | **98.67** | **98.15** | **98.25** | **90.74** |
| Random | 98.09 | 79.62 | 89.92 | 0.70 | 19.03 | 25.02 | 55.71 | 0.66 | 4.71 |
| AT | 96.76 | 93.91 | 64.56 | 1.96 | 51.30 | 84.38 | 45.60 | 1.76 | 16.44 |

Table 6: Performance (%) of shallow neural networks under PGD attack on CIFAR10.

| Method | Clean | $\epsilon = 0.01$ | | | | $\epsilon = 0.04$ | | | |
|--------|-------|-------------------|--|--|--|-------------------|--|--|--|
| | | $\text{PGD}_{5}^{un}$ | $\text{PGD}_{5}^{tar}$ | $\text{PGD}_{50}^{un}$ | $\text{PGD}_{50}^{tar}$ | $\text{PGD}_{5}^{un}$ | $\text{PGD}_{5}^{tar}$ | $\text{PGD}_{50}^{un}$ | $\text{PGD}_{50}^{tar}$ |
| SCE | 77.53 | 11.79 | 25.42 | 11.06 | 1.10 | 10.39 | 8.74 | 10.22 | 0.05 |
| Center | 76.46 | 11.10 | 18.69 | 10.97 | 0.35 | 10.13 | 4.52 | 6.40 | 0.12 |
| L-GM | 74.30 | 22.77 | 11.05 | 21.37 | 0.30 | 20.15 | 2.78 | 0.01 | 0.13 |
| MMC | 73.20 | 48.63 | 42.54 | 25.43 | 27.12 | 20.90 | 27.52 | 15.32 | 14.59 |
| I-SCE | **81.22** | **80.71** | **81.15** | **80.70** | **80.53** | **80.71** | **80.59** | **80.70** | **79.75** |
| Random | 78.71 | 15.23 | 24.53 | 9.78 | 1.37 | 10.41 | 9.70 | 10.20 | 0.21 |
| AT | 78.81 | 44.04 | 36.11 | 9.30 | 30.00 | 11.36 | 35.82 | 8.62 | 28.38 |

The above equation gives us that $h(f(x)) > 0 \iff e^{sf_i(x)+m} - e^{f_i(x)} > 0 \iff \frac{e^{sf_i(x)+m}}{e^{f_i(x)}} > 1 \iff e^{(s-1)f_i(x)+m} > 1 \iff (s-1)f_i(x) + m > 0 \iff s > \frac{f_i(x)-m}{f_i(x)}$. Hence, when the parameters $s$ and $m$ satisfy $s \geq 1 > \frac{f_i(x)-m}{f_i(x)}$ and $m > 0$, $h(f(x)) > 0$. When $s = 1$ and $m = 0$, $P_I(i|x)$ degenerates to $P(i|x)$.

Similarly, regarding $h(f(x+\delta))$, we have

$$
\begin{aligned}
h(f(x+\delta)) &= P_I(i|x+\delta) - P(i|x+\delta) \\
&= \frac{e^{sf_i(x+\delta)+m}}{e^{sf_i(x+\delta)+m} + \sum_{j \neq i} e^{f_j(x+\delta)}} - \frac{e^{f_i(x+\delta)}}{\sum_j e^{f_j(x+\delta)}} \\
&= \frac{\sum_{j \neq i} e^{f_j(x+\delta)}(e^{sf_i(x+\delta)+m} - e^{f_i(x+\delta)})}{(e^{sf_i(x+\delta)+m} + \sum_{j \neq i} e^{f_j(x+\delta)}) \sum_j e^{f_j(x+\delta)}}.
\end{aligned}
\tag{13}
$$

When $s \geq 1 > \frac{f_i(x+\delta)-m}{f_i(x+\delta)}$ and $m > 0$, $h(f(x+\delta)) > 0$.

Based on the above derivations, we calculate

$$
L_I - L = h(f(x)) - h(f(x+\delta))
\tag{14}
$$

To analyze the sign of the above equation, we compute the derivative of $h(f(x))$ with respective to $f_i(x)$ as

$$
\begin{aligned}
\frac{\partial h(f(x))}{\partial f_i(x)} &= \frac{\partial \left( \frac{e^{sf_i(x)+m}}{e^{sf_i(x)+m} + \sum_{j \neq i} e^{f_j(x)}} - \frac{e^{f_i(x)}}{\sum_j e^{f_j(x)}} \right)}{\partial f_i(x)} \\
&= \sum_{j \neq i} e^{f_j(x)} \frac{s e^{sf_i(x)+m}}{(e^{sf_i(x)+m} + \sum_{j \neq i} e^{f_j(x)})^2} - \sum_{j \neq i} e^{f_j(x)} \frac{e^{f_i(x)}}{(\sum_j e^{f_j(x)})^2} \\
&= \sum_{j \neq i} e^{f_j(x)} \left( \frac{s e^{sf_i(x)+m}}{(e^{sf_i(x)+m} + \sum_{j \neq i} e^{f_j(x)})^2} - \frac{e^{f_i(x)}}{(\sum_j e^{f_j(x)})^2} \right).
\end{aligned}
\tag{15}
$$

Table 7: Performance (%) of shallow neural networks under Deepfool attack.

| Method | MNIST | | | | | CIFAR10 | | | | |
| --- | --- | --- | --- | --- | --- | --- | --- | --- | --- | --- |
| | | $\epsilon = 0.01$ | | $\epsilon = 0.04$ | | | $\epsilon = 0.01$ | | $\epsilon = 0.04$ | |
| | Clean | $DF_{10}$ | $DF_{50}$ | $DF_{10}$ | $DF_{50}$ | Clean | $DF_{10}$ | $DF_{50}$ | $DF_{10}$ | $DF_{50}$ |
| SCE | 99.12 | 4.95 | 2.17 | 4.16 | 2.05 | 77.53 | 15.58 | 13.40 | 15.20 | 13.10 |
| Center | 98.17 | 26.98 | 17.63 | 24.49 | 16.10 | 76.46 | 6.40 | 1.45 | 6.32 | 1.43 |
| L-GM | 97.92 | 8.04 | 5.74 | 6.29 | 4.67 | 74.30 | 5.69 | 1.80 | 5.44 | 1.79 |
| MMC | 91.02 | 9.08 | 9.06 | 7.12 | 6.58 | 73.20 | 20.07 | 19.86 | 19.75 | 19.64 |
| I-SCE | 98.42 | **97.48** | **28.26** | **92.62** | **25.13** | **81.22** | **29.38** | **21.39** | **28.21** | **20.48** |
| Random | 98.09 | 71.46 | 42.60 | 59.34 | 49.49 | 78.71 | 28.75 | 21.54 | 31.75 | 25.44 |
| AT | 96.76 | 4.08 | 1.70 | 3.35 | 1.43 | 78.81 | 66.51 | 30.39 | 66.09 | 29.77 |

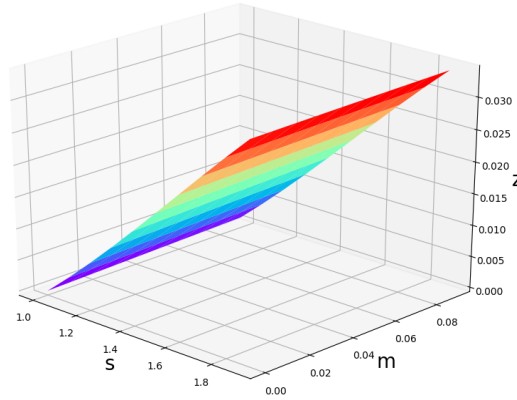

Figure 5: Parameter selection s,m

Considering that the perturbation $\delta$ is a just value that misleads the network, $f(x) > f(x+\delta)$. When $s$ and $m$ satisfy $\frac{se^{sf_i(x)+m}}{(e^{sf_i(x)+m}+\sum_{j\neq i}e^{f_j(x)})^2} - \frac{e^{f_i(x)}}{(\sum_j e^{f_j(x)})^2} > 0$, $h(f(x))$ is monotonically increasing. This guarantees that $L_I - L > 0$.

However, the condition $\frac{se^{sf_i(x)+m}}{(e^{sf_i(x)+m}+\sum_{j\neq i}e^{f_j(x)})^2} - \frac{e^{f_i(x)}}{(\sum_j e^{f_j(x)})^2} > 0$ is not easy to validate in the case of $s \geq 1$ and $m > 0$. Here, we conduct an experiment to empirically demonstrate its validation. Specifically, we compute the empirical values $\sum_{j\neq i} e^{f_j(x)} \approx 8, f_i(x) \approx 0.97$ by averaging the corresponding values of all samples in MNIST and CIFAR10. By employing these two values, we plot the 3D surface of $z = \frac{se^{sf_i(x)+m}}{(e^{sf_i(x)+m}+\sum_{j\neq i}e^{f_j(x)})^2} - \frac{e^{f_i(x)}}{(\sum_j e^{f_j(x)})^2}$ with respect to $s$ and $m$, which is shown in Figure 5. The surface indicates that $z$ is always larger than 0 when $s > 1$ and $m > 0$. This empirically demonstrates the validness of the conditions in the property.

