# OpenReview forum: "Improving robustness of softmax corss-entropy loss via inference information"
_ICLR.cc/2021/Conference — Reject_

### Official Review · AnonReviewer1 · 2020-10-28
**Official Blind Review**

**Rating:** 5
**Confidence:** 3

**Review:**

The authors present a new inference-SCE loss in order achieve a higher robustness to adversarial attacks.
They evaluate their model based on 2 datasets and compare their approach to several well known baselines.

The works seems convincing, even though I would have liked to see more extensive experiments on more complex dataset than CIFAR10 and MNIST.

---

> ### Author Response · Authors · 2020-11-24
> **Experiments on large-scale datasets are pending**
>
> Thanks for the valuable concern. We have followed the suggestion from the reviewer and are conducting a series of experiments on a large-scale dataset, Stanford Dogs. But due to the limited time of rebuttal, the results are not yet available by the deadline. We will complement the results in the final version if this paper is accepted.

---

### Official Review · AnonReviewer4 · 2020-10-28
**The proposed method is very similar to label smoothing but evaluates at different benchmarks.**

**Rating:** 4
**Confidence:** 4

**Review:**

Summary:
The paper claims that the vulnerability of DNN w.r.t adversarial samples comes from the overfitting issue when applying softmax cross-entropy loss. They propose I-SCE to mitigate the overfitting issue and show good performance in several adversarial attacks benchmarks.

The main concern from my point is the originality of this paper.  The author claims that avoiding overfitting can help improve the robustness under adaptive attacks. So the proposed method can be regarded as a regularization method to prevent overfitting issues. However, I think the proposed method shares a lot of similarity with label/model smoothing[1,2]. Also, the objective function eq8 basically is the same as Cosface loss function [3].

1) For eq. 1 and section 3.2, the authors state that the proposed method tends to produce the model prediction Q(y) close to P(y)-m instead of P(y). I think this is also the claim of label smoothing which avoids the model to output overconfident predictions w.r.t ground truth.
2) For eq.8, I would say the objective function is almost exactly the same as cosface loss ([3] eq.4) where s is the scale factor, m is the margin.  The cosface loss improves the model inter-class separability by setting a margin. I am not sure whether the proposed method benefits from inter-class separability or claimed label smoothing.

Overall, though the paper achieves good performance in several adaptive attacks benchmarks, I think the novelty of this paper is limited. Specifically, it shares a similar motivation and objective function to label smoothing and cosface. I hope the authors can elaborate on the difference between your work and label smoothing.

[1] Regularizing Neural Networks by Penalizing Confident Output Distributions, Gabriel Pereyra et al.
[2] Confidence Regularized Self-Training, Yang Zou et al.
[3] CosFace: Large Margin Cosine Loss for Deep Face Recognition, hao et al.

---

> ### Author Response · Authors · 2020-11-24
> **Softmax cross entropy loss based on large margin learning in free space**
>
> Thanks for the reviewer’s concern. The proposed method does share a similar motivation with label smoothing and large margin learning, yet with noticeable difference. The motivation discussed in the original submission is not well prepared and may confuse the readers and hence, we alternate our analyses of motivation in the revised version by clarifying the relations with label smoothing and large margin learning. Specifically, we identify that label smoothing could be viewed as a regularization of margin between the true label distribution and the output distribution, while the resultant model prefers a large margin between the features of different classes. Also considering the monotonical increasing property of softmax, the margin induced from label smoothing is similar to the large margin idea used in like arcface or cosface, at least in the perspective of the margin property of the well-trained models using label smoothing or a large margin-based loss. This is what we are trying to explain in the original submission, and is how the inference region comes.
>
> Next we explain the difference between our method and cosface. While the proposed method could be viewed as a margin-based loss, the difference to the arcface or cosface is how the margin (or inference information) is applied to the logits. Arcface or cosface normalizes the features and the weights such that the resultant features are located on a hyper-sphere, and the training process regresses the class targets along the surface of the hyper-sphere. Instead, the proposed method only normalizes the features but not the weights in order to locate the features in a free space, in which case the regression process can be performed in any direction. We advocate that freeing the sphere constraint will bring performance improvement of adversarial defencing, which is demonstrated in the experiments. The reason of the effectiveness may be that the adversarial perturbation causes a large variation in the feature space. Constraining the features on a hyper-sphere would bring a large feature shift if the normalization direction (to the sphere) is undesirable. By contrast, the proposed method prefers isotropic tolerance to feature perturbations, hence being better.

---

### Official Review · AnonReviewer3 · 2020-10-31
**review of "Improving robustness of softmax corss-entropy loss via inference information"**

**Rating:** 4
**Confidence:** 5

**Review:**

Paper proposes a modification to the widely adopted "Softmax Cross Entropy" (SCE) loss function that they refer to as I-SCE, with I standing for "Inference", that is designed for making the loss function more robust to adversarial examples.

The presentation of the idea is major mathematical flaws which make the idea authors are trying to present hard to grasp.

The main idea of the paper is to stretch the logit for the "true" class. This idea is presented in Eq. (7), where the logit term is stretched using the affine function x -> sx + m

This idea is introduced in the paper higher up, at Eq. (1) with adding a constant term to a probability! I had never seen a probability that is obtained from another one by adding a constant to the former. The reason being that adding a constant will break the "sum to one" property of a probability density function.  This is the first misleading / confusing math formula, which is used again in Eq. (3) & (4)

Another mathematical flaw in the paper is the text following Eq. (5): "This equation indicates that DKLI (P (y)∥Q(y)) − DKL(P (y)∥Q(y)) < 0, showing how inference KL divergence avoids the overfitting scenario.". I do not see why the difference of KL divergences being negative implies that overfitting is avoided. Perhaps authors can explain?

Figure 1 is not related to the text: why don't we have m but delta instead here? I don't get what this figure means

Figure 2 is misleading as well: which one is x, which one is y which is z axis, and what is presented here? can you add text to the axes so that I could understand what is going on there? how does this represent an adversarial attack anyway?

---

> ### Author Response · Authors · 2020-11-24
> **An interpretable and effective loss function to improve the adversarial defencing**
>
> Thanks for pointing out the ambiguous context of the paper. The analyses of the motivation originating from the KL divergence is somewhat less convincible, and what we are trying to explain is not very clear. Hence, in the revised version, we alternate our analyses of motivation by borrowing the ideas of label smoothing and large margin learning. Indeed, both label smoothing and large margin learning are two typical ways to improve the model robustness. We identify that label smoothing could be viewed as a regularization of margin between the true label distribution and the output distribution, while the resultant model prefers a large margin between the features of different classes. Also considering the monotonical increasing property of softmax, the margin induced from label smoothing is similar to the large margin idea used in like arcface or cosface, at least in the perspective of the margin property of the well-trained models using label smoothing or a large margin-based loss. This is what we are trying to explain in the original submission, and is how the inference region comes.
>
> While the proposed method could be viewed as a margin-based loss, the difference to the arcface or cosface is how the margin (or inference information) is applied to the logits. Arcface or cosface normalizes the features and the weights such that the resultant features are located on a hyper-sphere, and the training process regresses the class targets along the surface of the hyper-sphere. Instead, the proposed method only normalizes the features but not the weights in order to locate the features in a free space, in which case the regression process can be performed in any direction. We advocate that freeing the sphere constraint will bring performance improvement of adversarial defencing, which is demonstrated in the experiments. The reason of the effectiveness may be that the adversarial perturbation causes a large variation in the feature space. Constraining the features on a hyper-sphere would bring a large feature shift if the normalization direction (to the sphere) is undesirable. By contrast, the proposed method prefers isotropic tolerance to feature perturbations, hence being better.
>
> In Figure 1, delta implies an adversarial perturbation that is added to a clean input x. Hence, delta is not m. The grey and orange circle regions indicate where the samples are located. As seen, the grey region contains clean samples, while the orange region contains both clean samples and adversarial samples, indicating the distribution difference between clean and adversarial data. This figure depicts how the decision boundary is expected when using SCE loss and when using I-SCE loss.
>
> Figure 2 plots the performance of I-SCE under different settings of s and m, while each subfigure shows the results on different datasets and using different attacks. We have updated the figures in the revised version.

---

### Official Review · AnonReviewer2 · 2020-11-04
**Looks to me a degraded version of the adversarial training**

**Rating:** 5
**Confidence:** 4

**Review:**

The authors propose a loss function that is robust to the adversarial samples and claims the training with this loss function makes the model achieve better generalization ability.
However, there are a number of problems in this claim.

The method does not ground on a solid theoretical analysis\
I tried to interpret the analysis, but failed.
1. The right hand side of Eq(1) does not satisfy the rule of the probability. i.e., it does not satisfy $\sum_y P(y|x) = 1$ when $\sum_y P(y’|x) = 1$ and $m$ is a non-zero constant.
Thus the difference between two KLs around Eq.(5) does not make sense because $KL[P|Q]$ works as a valid divergence only when $P$ and $Q$ are probability distributions.
2. We cannot derive Eq.(3) from Eq.(1)
3.  If we add $m$ inside the exponential at the numerator for all the class $i$, it is equivalent not to add anything because it vanishes after normalization.
4. There is a lack of the explanation why the adversarial training is not sufficient. It seems for me adversarial training will be a natural choice if we want to make the model robust to the adversarial training.

Strange statement\
The authors wrote “Since the ground truth distribution of training data is **known**, $H(P(y))$ is a constant and hence,”
Usually we assume the ground truth distribution is **unknown**, if we know the ground truth distribution, there is no need to train the model. We can get immediately the minimizer of the loss. This may be a typo, but even if it is the mistake of “unknown”, still the statement is strange because it does not relate to whether $H(P(y))$ is a constant or not.

Experiment is not convincing\
The adversarial training works well for this experiment. Although the authors claim the adversarial training has a noticeable sacrifice of accuracy on clean examples. The mechanism is not well studied.
Also the authors tested their method only on MNIST and CIFAR10. It is not so appealing.

Reference\
The intuition of the authors claim has a relationship with the distributional robust learning. The authors should discuss the relationship with these methods.\
A. Sinha, et al., “Certifiable distributional robustness with principled adversarial training,” in ICLR 2018\
Amir et al., “Robustness to Adversarial Perturbations in Learning from Incomplete Data” in NeurIPS 2019.

-----------------------------------------------
I appreciate the authors to improve the clarity and consistency of the method.
I understand it has a practical value, easy to use and yet benefit from the method.
However, if the authors want to stress its practical value, we need more convincing experimental results beyond MNIST and CIFAR10.
Thus I concluded it is still below the acceptance threshold.

---

> ### Author Response · Authors · 2020-11-24
> **An interpretable and effective loss function to improve the adversarial defencing, not related to adversarial training.**
>
> Thanks for pointing out the ambiguous context of the paper. The analyses of the motivation originating from the KL divergence is somewhat less convincible, and what we are trying to explain is not very clear. Hence, in the revised version, we alternate our analyses of motivation by borrowing the ideas of label smoothing and large margin learning. Indeed, both label smoothing and large margin learning are two typical ways to improve the model robustness. We identify that label smoothing could be viewed as a regularization of margin between the true label distribution and the output distribution, while the resultant model prefers a large margin between the features of different classes. Also considering the monotonical increasing property of softmax, the margin induced from label smoothing is similar to the large margin idea used in like arcface or cosface, at least in the perspective of the margin property of the well-trained models using label smoothing or a large margin-based loss. This is what we are trying to explain in the original submission, and is how the inference region comes.
>
> While the proposed method could be viewed as a margin-based loss, the difference to the arcface or cosface is how the margin (or inference information) is applied to the logits. Arcface or cosface normalizes the features and the weights such that the resultant features are located on a hyper-sphere, and the training process regresses the class targets along the surface of the hyper-sphere. Instead, the proposed method only normalizes the features but not the weights in order to locate the features in a free space, in which case the regression process can be performed in any direction. We advocate that freeing the sphere constraint will bring performance improvement of adversarial defencing, which is demonstrated in the experiments. The reason of the effectiveness may be that the adversarial perturbation causes a large variation in the feature space. Constraining the features on a hyper-sphere would bring a large feature shift if the normalization direction (to the sphere) is undesirable. By contrast, the proposed method prefers isotropic tolerance to feature perturbations, hence being better.
>
> To answer the question about adversarial training, we consider that adversarial training is not an ultimate solution to adversarial defencing because the distribution of adversarial examples available in training is biased with respect to the attacking manners used in real applications. Unless we could generate all kinds of adversarial examples, the robustness of the adversarial model could be restricted. In contrast, the proposed method does not rely on the number of adversarial examples, but on the characteristics of the solution space. That is, the proposed method favours a solution that poses large margins between different classes, in which case the perturbation hardly drives the samples across the decision boundary.
>
> The datasets used in experiments follow the settings of existing adversarial defencing papers. Here, we follow the suggestion of the reviewer and are conducting a series of experiments on a large-scale dataset, Stanford Dogs. But due to the limited time of rebuttal, the results are not yet available by the deadline. We will complement the results in the final version if this paper is accepted.
>
> Regarding the references pointed by the reviewer, we identify that those methods characterize the adversarial perturbation (or specifically, the worst disturbance to the distribution of clean data) by using the Wasserstein divergence, where the Wasserstein ball has a good property of modelling robustness of unknown data. It is theoretically guaranteed that the resultant optimization problem is strong convex, hence being computationally efficient. Instead, we address the adversarial defencing problem from a different viewpoint, which produces an easy-to-implement, plugin-in, interpretable, and effective loss function that yields improved performance.

---

### Decision · Program_Chairs · 2021-01-07
**Final Decision**

**Decision:**

Reject

**Comment:**

This paper presents an inference-softmax cross entropy (I-SCE) loss, a modification to the widely adopted "Softmax Cross Entropy" (SCE) loss, to achieve better robustness against adversarial attacks. The original submission had critical issues on motivation, theoretical analysis and experiments. Although the authors provided a revised version, it needs another round of thorough examination before publishing.